# Pathological and Pharmacological Roles of Mitochondrial Reactive Oxygen Species in Malignant Neoplasms: Therapies Involving Chemical Compounds, Natural Products, and Photosensitizers

**DOI:** 10.3390/molecules25225252

**Published:** 2020-11-11

**Authors:** Yasuyoshi Miyata, Yuta Mukae, Junki Harada, Tsuyoshi Matsuda, Kensuke Mitsunari, Tomohiro Matsuo, Kojiro Ohba, Hideki Sakai

**Affiliations:** Department of Urology, Nagasaki University Graduate School of Biomedical Sciences, 1-7-1 Sakamoto, Nagasaki 852-8501, Japan; ytmk_n2@yahoo.co.jp (Y.M.); harada-junki@nagasaki-u.ac.jp (J.H.); matsudatsuyoshi9251@gmail.com (T.M.); ken.mitsunari@gmail.com (K.M.); tomozo1228@hotmail.com (T.M.); ohba-k@nagasaki-u.ac.jp (K.O.); hsakai@nagasaki-u.ac.jp (H.S.)

**Keywords:** oxidative stress, mitochondrial reactive oxygen species, chemical compounds, natural product, photodynamic therapy, malignancies

## Abstract

Oxidative stress plays an important role in cellular processes. Consequently, oxidative stress also affects etiology, progression, and response to therapeutics in various pathological conditions including malignant tumors. Oxidative stress and associated outcomes are often brought about by excessive generation of reactive oxygen species (ROS). Accumulation of ROS occurs due to dysregulation of homeostasis in an otherwise strictly controlled physiological condition. In fact, intracellular ROS levels are closely associated with the pathological status and outcome of numerous diseases. Notably, mitochondria are recognized as the critical regulator and primary source of ROS. Damage to mitochondria increases mitochondrial ROS (mROS) production, which leads to an increased level of total intracellular ROS. However, intracellular ROS level may not always reflect mROS levels, as ROS is not only produced by mitochondria but also by other organelles such as endoplasmic reticulum and peroxisomes. Thus, an evaluation of mROS would help us to recognize the biological and pathological characteristics and predictive markers of malignant tumors and develop efficient treatment strategies. In this review, we describe the pathological significance of mROS in malignant neoplasms. In particular, we show the association of mROS-related signaling in the molecular mechanisms of chemically synthesized and natural chemotherapeutic agents and photodynamic therapy.

## 1. Introduction

Reactive oxygen species (ROS) are intracellular signaling molecules formed by the reduction of O_2_, and include superoxide anion (O_2_^−^), hydrogen peroxide (H_2_O_2_), as well as hydroxyl radicals (OH•) [1]. Under normal physiological conditions, this substance participates in the maintenance of metabolic homeostasis, and exhibits regulatory roles in proliferation and differentiation [2,3,4]. During the last two decades, numerous in vivo and in vitro studies have supported the opinion that dysregulation of ROS plays an important role in the etiology and pathology of diseases such as vascular disease, diabetes, and malignant tumors by damaging cellular components, and in the pharmacological mechanisms of therapeutics [5,6,7,8,9].

Mitochondria are well known as dynamic cellular organelles involved in bioenergetics and metabolic signaling [10]. They are also recognized as a major source of ROS, and mitochondria-derived ROS (mROS) are tightly regulated in the cell. Mitochondrial ROS (mROS) play crucial roles in various biological activities including cell differentiation, survival, and immunity, and can accumulate upon mitochondrial dysfunction, imbalance of antioxidant homeostasis, and/or hypoxic conditions [5,10,11,12]. Nicotinamide adenine dinucleotide phosphate (NADPH) oxidases (NOXs) are also a major source of ROS [12]. The NOX family encompasses seven homologs, NOX1–NOX5, DUOX1, and DUOX2, which are recognized as multi-subunit electron transporting membrane proteins [12,13]. Initially, NOXs were speculated to occur in phagocytes, however, they were subsequently detected in almost all tissues and in various cellular components including the cellular membrane, nucleus, endoplasmic reticulum, and mitochondria [14,15]. Endoplasmic reticulum produces ROS by NOX-independent pathways; phagosomes, peroxisomes, and Golgi apparatus are also reported to be sources of intracellular ROS [15,16,17]. Crosstalk between NOXs and mitochondria, and subsequent intracellular ROS production, is essential to various biological activities such as tissues repair and angiogenesis [18].

There is a general agreement that mitochondria and NOXs are the two major sources of intracellular ROS production under pathological conditions. However, we should note that several molecules are associated with intracellular ROS production by complex mechanisms. For example, forkhead box O transcription factors (FOXOs), which comprise a family of context-dependent transcription factors, are known to maintain intracellular ROS balance via activation of the PI3K/Akt pathway [19,20,21]. Furthermore, enzymes such as xanthine oxidase, nitric oxide synthase (NOS), lipoxygenase, cyclooxygenase, monoamine oxidase, and cytochrome P450 are regulators of intracellular ROS production [18,22,23,24,25].

In this review, we emphasize that, although it is true that mitochondria are one of the major and most representative sites of ROS production, all intracellular ROS is not always supplied from mitochondria (Figure 1).

In fact, a previous report on astrocytes showed that the relative ratio of subcellular ROS levels in the cytosol, nucleus, and mitochondria was approximately 0.29:0.3:1 [26]. In a strict sense, biological and pathological roles of mROS are different from that of cytosolic ROS, and mROS production does not always reflect intracellular ROS status [27]. For instance, regulation of hypoxia-inducible factor (HIF) pathway under hypoxia is one of the best characterized roles of mROS, because cells are unable to produce mROS and stabilize HIF-1α upon mitochondrial DNA depletion [28,29,30]. Studies have indicated that lipopolysaccharides stimulated both intracellular ROS and mROS in gastric cancer cells, and N-acetyl-L-cysteine (NAC; an antioxidant) and diphenylene iodonium (DPI, NADPH oxidase inhibitor) inhibited intracellular ROS production [31]. However, this study also showed that DPI inhibited production of mROS, whereas NAC did not [31]. Thus, production of intracellular ROS and mROS are regulated by different mechanisms. Mitochondrial ROS and intracellular ROS production, induced by a variety of anticancer agents, varied under hypoglycemic conditions relative to normal conditions in cancer cells [32]. In addition, there is the opinion that inhibition of mROS is more effective than that of intracellular ROS under pathological conditions [33]. In fact, mROS production and antioxidant systems of mitochondria are promising targets of cancer therapy [3,30,34]. Antioxidative mechanisms in mitochondria depend on the types of oxidative stress and target organs, and these mechanisms are not always similar to other organelles [35,36].

There are numerous systematic reviews that have described the detailed pathological significance of ROS in malignant cells, as well as their pharmacological impact in chemotherapy [8,20,37]. However, until now, the role of mROS in malignant tumors, especially in cancer treatment with natural products and photodynamic therapy (PDT), has not been reviewed. Therefore, the main aim of this review is to describe the specific roles of mROS in cancer treatments with chemical compounds, natural products and their extracts, and photosensitizers. As per previous studies, mROS production is evaluated using three methods, mitochondrial-specific probe, MitoSOX, and dihydrorhodamine 123, and it is measured in isolated mitochondria [27,38,39]. It is not pertinent to distinguish between mROS and intracellular ROS production, because mROS levels are speculated to be similar to intracellular ROS levels in physiological and pathological conditions. However, we believe that deciphering the specific roles and regulative mechanisms of mROS production is essential to develop patient-specific treatment strategies for cancer.

## 2. Chemical Compounds

Chemotherapy is a conventional cancer therapy, particularly in advanced or metastatic stages. It is well known that ROS production by chemotherapeutic agents mediates anticancer effects. Chemotherapeutic agent-induced ROS production overwhelms the cells’ oxidative power and antioxidant defense. In addition to approved anticancer agents, several newly developed anticancer agents incorporate ROS generating compounds to amplify oxidative stress in cancer cells, and subsequently suppress tumor growth [40,41]. Contrary to expectation, studies regarding pharmacological roles of mROS in chemical compound-induced anticancer effects are limited.

### 2.1. Platinum-Based Chemotherapeutic Agents

Platinum compounds such as cisplatin (CDDP) and carboplatin are the most important and effective chemotherapeutic agents for several types of cancers [42,43,44]. The underlying molecular mechanisms of platinum compounds against cancer cells involves various cancer-related molecules and factors, and mitochondrial damage is one of most important means of tumor growth inhibition [43,45]. In fact, in head and neck squamous cell carcinoma, CDDP-induced cancer cell apoptosis occurs via direct action on mitochondrial DNA and is not dependent on nuclear DNA [45]. Numerous studies have shown that ROS production was recognized as a key step of platinum agent-induced anticancer effects [37,46]. Platinum agents have also been shown to damage cancer cells via mitochondrial dysfunction [47,48]. However, the association between platinum agents-induced anticancer effects and mROS is not fully understood.

A study reported that when non-small cell lung cancer cells (A549 cells) and prostate cancer cells (DU145 cells) were continuously exposed to CDDP at an IC_50_ dose for 24 h, intracellular ROS levels were significantly increased as compared with the base line (0 h) after 16 and 24 h, respectively [49]. Similar observations were reported for mROS levels in non-small cell lung cancer cells [49]. Although similar changes in intracellular ROS and mROS were observed after 24 h of CDDP treatment in prostate cancer cells, the results varied after 16 h [49]. Mitochondrial ROS levels in prostate cancer cells after CDDP exposure for 16 h was significantly increased compared to the initial level, whereas a significant change was not found in intracellular ROS levels [49]. Thus, mROS production increased at an earlier time point by the treatment as compared with intracellular ROS. This study concluded that mitochondria were a major source of CDDP-induced ROS generation in these cancer cells, and this mitochondria-mediated process is a major component of CDDP-induced cytotoxicity, in addition to nuclear DNA damage [49]. In recent years, CDDP has been reported to increase mROS production in ovarian cancer cell lines. mROS levels in CDDP-sensitive ovarian cancer cells (OVCAR-3, OVCAR-4, and IGROV-1) were higher relative to less sensitive or resistant cell lines (OVCAR-5, OVCAR-8, and A2780) [43]. These findings support the opinion that a part of the anticancer effects of CDDP is caused by enhanced mROS production. However, the biological and pharmacological roles of mROS in platinum agents-based chemotherapy are not fully understood.

Several factors are associated with the anticancer effects of chemotherapeutic agents via regulation of mROS production. For example, the presence of aconitase 2 (ACO2), which is located in the mitochondrial matrix and plays an important role in cellular metabolism, has been reported to activate CDDP-induced cell death via p53 signaling pathway in breast cancer cells, with mROS production being closely associated with this mechanism (MCF-7 cells) [50]. Interestingly, this study also showed that enhanced mROS production was regulated by CDDP and ACO2, and cytoplasmic ROS production was associated with the p53-mediated apoptotic pathway [50]. Thus, cancer cell death caused by CDDP is regulated by complex mechanisms involving mROS and intracellular ROS production. Elucidation of this mechanism is important to understand the anticancer effects and safety of CDDP-based chemotherapy.

### 2.2. Taxane

Paclitaxel is a member of the taxane group of chemotherapeutic agents that induce cell death via inhibition of microtubules in cancer cells. It is part of the standard regimen and clinical trials in various types of cancers [44,51,52]. The primary source of paclitaxel-induced ROS production has been identified to be mitochondria [53]. The same study also showed that a mitochondrial uncoupler (carbonyl cyanide p-(trifluoromethoxy)-phenylhydrazone) suppressed paclitaxel-induced mROS production in lung cancer cell lines [53]. Interestingly, high expression of uncoupling protein (UCP)-2 in lung cancer cells under oxidative stress stimulated mitochondrial uncoupling and decreased ROS production [53]. Finally, the authors concluded that UCP-2 and ROS were useful anticancer therapeutic targets in lung cancer [53].

Docetaxel is also a member of the taxane class of anticancer drugs. Docetaxel alone or docetaxel-based regimens are employed in standard chemotherapy for various types of cancers [54,55]. A study showed that prolonged exposure to docetaxel and 5-fluorouracil (5-FU) in colon cancer cells (HT29 and LOVO cell lines) under hyperglycemia led to decreased anticancer effects of these drugs, and suppression of mROS production [32]. Mitochondrial ROS levels were significantly decreased after treatment of 5 and 10 μM docetaxel for 24 h and 25 and 50 μM 5-FU for 72 h in colon cancer cells cultured in hyperglycemic medium; however, a significant change was not observed in control cells [32]. These findings explain the reason why anticancer effects of chemotherapy are often weak in patients with diabetic mellitus. This study showed that appropriate control of serum sugar levels improves anticancer effects via upregulation of mROS production in cancer cells in patients with diabetic mellitus.

### 2.3. Other Conventional Chemotherapeutic Agents

Uncoupling protein 2 (UCP2) is a mitochondrial transporter protein that is closely associated with energy homeostasis [56,57]. This substance is known to be expressed in various types of cancer cells and suppresses mROS production [58,59,60]. Interestingly, there was a report that chemosensitivity for gemcitabine was increased when UCP2 was knocked down in gall bladder cancer cells [60]. This study showed that UCP2 knockdown enhanced mROS production in gemcitabine-exposed gallbladder cancer cells [60]. Increased mROS production by gemcitabine may be involved in its anticancer effect in gallbladder cancer. Gemcitabine is recognized as a key chemotherapeutic agent and is effective in a variety of cancers, such as urothelial cancer and pancreatic cancer [61,62].

Temozolomide is an oral alkylating agent, and temozolomide-based regimens are used for various types of malignant tumors including malignant glioma [63,64]. A recent study showed that superoxide dismutase (SOD) 2 promoted chemoresistance to temozolomide in glioblastoma cells (U87MG and A712 cells) and temozolomide resistance relied upon tight regulation of mROS production and enrichment of tumor-infiltrating cells [65]. We are in agreement that SOD2 may be a potential target for treatment strategies for glioblastoma. Furthermore, we also suggest that anticancer effects can be enhanced by adequately controlling mROS production.

### 2.4. TRAIL-Related Chemical Compounds

Tumor necrosis factor-related apoptosis-inducing ligand (TRAIL) is a useful potential therapeutic target in malignant tumors because this substance induces cell death in various malignant cells, but not in normal cells [66]. Certain malignant cells are resistant to TRAIL-induced cell death [67]. Many investigators pay special attention to treatment strategies based on the regulation of TRAIL-related processes [68]. Several metabolic inhibitors of the mitochondria, such as complex I inhibitor (rotenone (ROT)), complex III inhibitor (antimycin A (AM)), and mitochondrial uncoupling agent (carbonyl cyanide p-(trifluoromethoxy) phenylhydrazone (FCCP)) have been reported to increase mROS levels, and subsequently stimulated TRAIL-induced apoptosis in human leukemia cells (Jurkat cells) [69]. Similar findings of increased mROS levels attributed to ROT, AM, and FCCP, and enhanced TRAIL-induced apoptosis have been reported in human melanoma cells (A375 cells) [66]. This study also showed that the capability of mROS production was highest in AM (7.8-fold versus control) treatment as compared with that in FCCP (2.1-fold) and ROT treatment (1.9-fold) [66]. These reports suggested that increased mROS production via mitochondrial dysfunction enhanced TRAIL-induced proapoptotic processes [66,69].

Reactive oxygen species modulator 1 (Romo 1) and plasma-activated medium (PAM) are regulators of mROS production and TRAIL-induced apoptosis in malignant cells [70,71]. Plasma-activated medium (PAM), a solution irradiated with nonthermal atmospheric pressure plasma, has exhibited anticancer effects via regulation of ROS production in several cancers [72]. PAM has induced TRAIL-induced apoptosis by increasing mROS level in human cervical cancer cells (Hela cells) [70]. A recent study showed that Romo1, a mitochondrial inner membrane channel protein, regulated mROS production and suppressed TRAIL-induced apoptosis in colon cancer [71]. Inhibition of Romo 1 function led to enhanced TRAIL-induced apoptosis. Thus, this substance may be an effective therapeutic target for TRAIL-based therapy and contributes to improving the development of novel therapeutic agents for colorectal cancer [71].

### 2.5. Other Chemical Compounds

C/EBP homologous protein (CHOP) belongs to the family of CCAAT/enhancer binding proteins (C/EBPs) and is associated with cell differentiation, proliferation, and energy metabolism [73]. It is well known that CHOP plays important roles in endoplasmic reticulum stress-induced apoptosis [73,74]. Several reports have shown that activation of CHOP-related pathways led to enhanced cancer cell apoptosis [75,76]. Therefore, CHOP is a potential therapeutic target in cancer treatment. LGH00168 was developed as a CHOP activator and has been reported to exhibit anticancer effects via mROS-related mechanisms in various types of cancer, particularly lung cancer [76]. Briefly, in vitro experiments showed that LGH00168 inhibited cancer cell proliferation in ovarian, hepatoma, cervical, breast, colon, and lung cancer cells, and suppressed tumor growth in lung tumor (A549 cells) xenograft bearing mice via dose-dependent increase of mROS [76]. Furthermore, this study showed that severe endoplasmic reticulum stress, NF-κB inhibition, and dysregulation of mitochondria were associated with such mROS-mediated cell death by this substance [76].

Biguanides, composed of metformin and phenformin, possess hypoglycemic functions and are employed for treating type 2 diabetes [77]. Metformin has been commonly used for treatment of patients with diabetic mellitus worldwide because of higher values of clinical efficacy, safety, and cost [78]. In addition to antidiabetic functions, biguanides have been reported to possess antineoplastic effects in various types of malignant tumors, such as cervical cancer, breast cancer, and pancreatic neuroendocrine tumors [79,80,81]. The associated anticancer mechanisms involve cell signaling molecules such as FAK, Akt, Rac1, RhoA, cyclin D1, and proliferating cell nuclear antigen [79,81]. There was a report implicating biguanides in the modulation of the metabolic profile of malignant lymphocytes, wherein mROS and activation of HIF-1α played crucial roles [82]. This study showed that biguanides suppress tumor growth in a xenograft model of human leukemia [82]. Biguanide-induced metabolic rewiring, though enhanced ROS production and increased HIF-1α activation, led to antineoplastic effects. Interestingly, combinatorial treatment with phenformin and HIF-1α inhibitor elicited a greater suppressive effect on xenograft tumor growth (PX-478) as compared with monotherapy with phenformin or PX-478 [82]. The combination of biguanides and HIF-1α inhibitors may be an effective treatment strategy for lymphoid leukemia [82]. We have summarized anticancer chemical compounds affecting mROS production in Table 1.

## 3. Natural Products

Natural products are widely used as health supplements and drugs worldwide, particularly in Asia. They are safe and relatively inexpensive, and investigators support the opinion that natural products exhibit anticancer effects and improve prognosis in various types of cancers [84,85,86,87]. In recent years, increased risk of toxicity and reduction of antioxidant activity by uncontrolled and excessive consumption of natural products and their extracts has become problematic [86]. Therefore, it is essential to understand the detailed molecular mechanisms of their biological activities for application of natural products and their extracts in cancer therapy. Regulation of ROS production is one of the important mechanisms in the pharmacological application of natural products [86,88]. However, most studies have discussed the relationship between ROS production and anticancer effects in terms of intracellular ROS, and not mROS. In this section, we review the biological significance of mROS in anticancer therapies involving natural products and their extracts.

### 3.1. Matairesinol

Matairesinol, a dibenzylbutyrolactone plant lignan, is present in a variety of foods such as oil seeds, whole grains, vegetables, and fruits [89]. It possesses anti-inflammatory, antiestrogenic, immunosuppressive, and anticancer activities [90,91,92,93]. Several reports have shown that matairesinol served as an antioxidant both in vivo and in vitro [94,95]. A study indicated that matairesinol suppressed proliferation of cervical cancer cells (Hela cells) [96]. The authors also showed that matairesinol suppressed angiogenic processes such as cell proliferation, invasion, and tube formation of human umbilical vascular endothelial cells (HUVECs) cultured in tumor conditioned medium [96]. Importantly, matairesinol treatment induced the inhibition of mROS production, suppression of HIF-1αstabilization, and decreased expression of vascular endothelial growth factor (VEGF) [96]. Thus, matairesinol is speculated to be a potential anticancer agent that targets mROS-related systems.

### 3.2. Pancratistatin

Pancratistatin is a natural compound isolated from spider lily, and it exhibits cytotoxic effects and pro-apoptotic activity against a variety of malignant tumors such as melanoma, colon cancer, and breast cancer [97,98,99]. Although its detailed molecular mechanisms of cytotoxicity are not fully understood, several investigators have reported that mitochondria were the main target of pancratistatin [97,99,100]. McLachlan et al. reported that increased ROS and loss of mitochondrial membrane potential played an important role in pancratistatin-related anticancer activities in several cancer cells [100]. JC-TH-acetate-4 (JCTH-4), a C-1 acetoxymethyl analogue of 7-deoxypancratistatin, is a synthetic compound that exhibits an anticancer activity similar to that of pancratistatin [101]. JCTH alone, and in combination with the natural compound curcumin, have induced apoptosis in osteosarcoma cell lines (U-2 OS and Saos-2 cells), however, this effect was not observed in normal cells (normal human fetal fibroblasts (NFF) cells and normal human osteoblast, and Hob cells). JCTH-4 increased ROS generation in isolated mitochondria of osteosarcoma cells [102].

### 3.3. Betulin

Betulin (3-lup-20(29)-ene-3*β*,28-diol), a pentacyclic lupine-type triterpenoid, which has been found in the bark of birch trees, has been reported to exhibit cytotoxic and antigrowth effects in various types of malignant tumors [34,103,104]. However, such anticancer effects were mainly detected in cancer cell lines, and not in vivo models. The limited effect of botulin may be attributed to poor solubility due to its lipophilic structure [34]. 28-O-α-l-rhamnopyranosylbetulin 3β-O-α-l-rhamnopyranoside (Bi-L-RhamBet), a compound synthesized from betulin, suppressed tumor growth of lung cancer in both in vivo and in vitro studies performed on LLC1 tumor-bearing mouse and lung cancer cell lines [34]. Bi-L-RhamBet exhibited its cytotoxic activity toward cancer cells by inhibition of cell cycle, induction of apoptosis, caspase activation, and DNA fragmentation, whereby mROS production was involved in caspase activation and apoptosis [34].

### 3.4. Tannic Acid

Tannic acid (TA), a natural product containing polyphenol, is present in various food items such as fruits and vegetables [105]. This substance is multifunctional as it demonstrates antiviral and antibacterial activities and preventive effects for vascular diseases [106,107]. Recent studies have shown that TA was useful for cancer prevention and treatment [108,109]. For example, it has been reported to induce extrinsic apoptosis in human embryonic carcinoma cells [110]. Interesting, TA-induced apoptosis was regulated by two different pathways; one pathway was mediated via Wnt/β-catenin signaling and the other pathway involved mROS production [110]. This study showed that mROS production increased in a TA concentration-dependent manner, and TRAIL-mediated extrinsic apoptosis pathway was activated in response to mROS generation [110].

### 3.5. Curcumin

Curcumin is recognized as a bioactive compound that belongs to the family of curcuminoids, which are yellow pigments extracted from turmeric rhizomes. This substance has been reported to have exhibited anticancer effects in various malignant cells [87,111]. Curcumin and its analogs enhance the anticancer effects of conventional chemotherapeutic agents [112,113]. ROS production is one of the molecular mechanisms of curcumin-induced anticancer activity [114,115]. However, the pharmacological role of mROS in curcumin-related anticancer mechanisms is not fully understood. The biological role of mROS in the anticancer effects of curcumin analog, ALZ003, in glioblastoma cells has been elucidated. This report showed that ALZ003 suppressed the growth of glioblastoma by decreasing the AR protein level, and ALZ003 stimulated oxidative stress via inhibition of glutathione peroxidase 4. Enhanced mROS production was also a part of the tumor-suppressive mechanisms of ALZ003 in glioblastoma cells [116].

### 3.6. Vitamin C

Vitamin C, also known as ascorbic acid, is an essential dietary requirement and is detected in a wide variety of vegetables and fruits. This substance supplements are widely consumed for nutritive purposes as it is known to exhibit antioxidative and free radical-scavenging functions, and it is also associated with tumor characteristics, quality of life, and prognosis in cancer patients [117,118,119,120]. It also induces the degradation of HIF-1 and activates the immune system via NK and T cells and monocytes [121,122]. Furthermore, ROS production has been reported to be associated with vitamin C-related biological activities and anticancer effects [118,120]. One report described the pharmacological effect of mROS by vitamin C in embryonic carcinoma [33]. In the F9 embryonic carcinoma cell line, vitamin C suppressed mROS production and cell apoptosis via a sirtuin1-SOD dependent mechanism [33]. The methodology employed in this study was as follows: First, F9 cells were established from a clonal mouse teratocarcinoma (not from human cells) and, second, mitochondrial oxidative stress was induced by sodium fluoride (NaF). Notably, this cell line has been commonly used to analyze the molecular mechanisms of malignant behavior [123]. It is possible that NaF negatively influences physiological conditions such as bone metabolism, immune response, fertility, and testicular function [124,125,126,127].

## 4. Photodynamic Therapy

Dysregulation of mitochondrial function by various internal and external stimuli leads to suppression of energy supply and induction of apoptosis. UV radiation and laser irradiation are the most representative external stimuli in such pathological processes [128,129]. UV and laser irradiation promote oxidative stress by enhancing ROS production [130,131]. Many investigators support the findings that UV and laser irradiation affect biological and pathological conditions such as skin damage, aging of the eye, and wound healing via regulation of ROS production and mitochondrial function [132,133,134]. Mitochondria-mediated apoptosis is an important determinant for tumor progression and successful treatment of various cancers [135,136]. Furthermore, mROS play crucial roles in mitochondrial-mediated apoptosis [137,138]. In breast cancer cells, mROS production was elevated upon exposure to UV irradiation, which subsequently led to UV irradiation-induced apoptosis [139].

Mitochondria are considered to be suitable targets for photodynamic therapy (PDT) because they are sensitive to light [26,140]. PDT is regarded as a minimally invasive therapeutic tool which incorporates photosensitizers and light of a specific wavelength. PDT increases endogenous ROS production via regulation of mitochondrial membrane potential [141]. PDT has a beneficial impact on various pathological conditions. In fact, PDT is an effective and promising therapy for malignant tumors, skin diseases, and periodontitis [142,143,144]. In cancer treatment, PDT is often performed with surgery and/or chemotherapy [145,146,147]. Thus, understanding the biological roles of photosensitizers at the molecular level is essential to develop PDT-based treatment strategies. In this section, we discuss the pathological and pharmacological role of mROS in cancer treatment with PDT.

### 4.1. Photosensitizers and mROS

Since the 1990s, photosensitizers have been widely used for the treatment of advanced cancer. In 1995, porfimer sodium was approved by the U.S. Food and Drug Administration for PDT-based treatment of obstructive esophageal cancer, and its clinical efficacy was appropriate [148]. Subsequently, porfimer sodium is now commonly used for lung cancer and endobronchial cancer [149]. Several studies have implicated mROS generation to be involved in porfimer sodium-induced anticancer effects [136,150]; however, most studies indicated that porfimer sodium affected intracellular ROS levels [136,151,152]. There is no strong evidence that mROS mediates the therapeutic effects of porfimer sodium. Thus, first-generation photosensitizers do not seem to specifically target the mitochondria [149].

Second-generation photosensitizers have been developed with high purity, long wavelength absorption, photosensitivity, and tissue selectivity. Motexafin lutetium, temoporfin, palladium bacteriopheophorbide, purpurins, verteporfin, and protoporphyrin IX precursors are second-generation photosensitizers designed with mitochondria-specific targets [149]. Therefore, regulation of mROS production is one of the main mechanisms of action of second-generation photosensitizers. PDT with verteporfin and 488 nm laser irradiation (benzoporphyrin-derivative monoacid ring A (BPD-MA), i.e., verteporfin) has induced mitochondrial swelling and mROS production and resulted in apoptosis in glioma cells [153]. Similarly, other investigators have shown that verteporfin with 690 nm laser irradiation induced specific and severe damage to the mitochondrial membrane in astrocytes with elevated mROS generation [26]. PDT with verteporfin has been effective in various types of cancers such as gastric cancer, rectal cancer, and breast cancer [154,155,156]. Intracellular ROS production plays a crucial role in the anticancer effects of second-generation photosensitizers and its safety has been investigated in various types of malignant tumors [149,157,158,159,160]. Unfortunately, the specific roles of mROS in PDT with second-generation photosensitizers, other than verteporfin, are still not clear.

We summarize anticancer natural products and photosensitizer that affect mROS production in Table 2.

### 4.2. 5-Aminolevulinic Acid-Mediated Photodynamic Therapy and mROS

At present, 5-aminolevulinic acid (5-ALA)-mediated PDT is recognized as a minimally invasive therapeutic modality for various malignant tumors [161,162,163]. There is general agreement that ROS production is essential to elicit the therapeutic effects of 5-ALA-mediated PDT [164]. Importantly, 5-ALA is produced and located in the mitochondria; therefore, exogenously supplemented 5-ALA can modulate mitochondrial functions [165]. PDT with 5-ALA reduced mROS levels and, subsequently, the mROS-dependent autophagy pathway was suppressed in keloids [164]. This study also showed that the biological effects of 5-ALA-based PDT were regulated by sirtuin (SIRT) 1, SIRT3, and mitochondrial oxidative scavenger, superoxide dismutase 2 (SOD2), which are closely associated with mROS homeostasis [33,164,166]. The detailed pharmacological roles of mROS in 5-ALA-based PDT in malignant tumors are not fully understood, although mitochondria are known to be the main target. Therefore, we suggest that regulation of mROS is a potential means to further improve the efficacy of 5-ALA-based PDT in cancer patients. However, we emphasize that further studies on the pharmacological effects of mROS in 5-ALA-mediated PDT for malignant tumors are warranted to improve its efficacy.

Finally, a schema of molecular mechanisms of mROS-related anticancer effects is showed in Figure 2.

## 5. Future Perspectives on the Pharmacological Roles of mROS in Cancer Treatments

### 5.1. Mitochondrial ROS in Immune Therapy

Presently, immune checkpoint inhibitors are commonly used for cancer therapy in several cancers, and ROS is closely associated with the tumor microenvironment and immunity [9]. ROS accumulation has been suggested to be associated with the anticancer effects of these compounds, including induction of cell death and prolongation of survival periods in gastric cancer and lung cancer [167,168]. There is an excellent review about the regulation of programmed-death ligand 1 (PD-L1) expression by ROS in cancer cells [169]. In this review, the authors analyzed 15 pharmacological ROS modulators, and indicated that ROS effectors regulated PD-L1 expression in cancer cells [169]. Importantly, this review also concluded that further studies were necessary to delineate the complex crosstalk between ROS and PD-L1 in the tumor microenvironment [169]. It is evident that drug-induced alteration of ROS equilibrium in cells can significantly dysregulate immune checkpoints, however, the pathological roles and pharmacological role of mROS or mROS-modulating agents are not fully understood.

### 5.2. Antioxidants and mROS

In this review, we focused on the role of mROS in malignant tumors. However, we should note that a balance between oxidative stress mediated by ROS and the antioxidant system is a control point for the biological and pathological state of cells, tissues, and organs [25,170]. Although excessive ROS stimulates malignant behavior in cells, prolonged exposure to ROS leads to cancer cell death if not neutralized by antioxidants [3,30,171]. Several antioxidants, including NAC and dithioerythritol (DTE), suppress aggressiveness of malignant tumors in a breast cancer cell line (MDA-MB-231 cells) [172]. Deep insight into the complex and delicate redox balance of a cell is essential for understanding the pathological significance and efficacy of mROS as a therapeutic target for malignant tumors. Unfortunately, we were not able to discuss details of mitochondrial antioxidant systems and the mechanisms of redox in this review because of extensive studies regarding this topic in literature. Moreover, excellent reviews about tumor-specific generation of ROS, mROS, and redox-based targeted therapeutic strategies have been published previously [3,173].

Clinical trials have been performed to verify the anticancer effect and safety of various antioxidants in patients with malignant tumors. However, there was no direct evidence that antioxidant drugs and supplements improved the outcome or had beneficial effects [174]. On the contrary, clinical trials showed that a variety of antioxidants, including β-carotene, vitamin A, and vitamin E, seemed to increase the risk of cancer-related mortality [174]. Detrimental effects of antioxidants were also observed in several animal cancer models [175,176]. Administration of antioxidants, NAC, and vitamin E enhanced tumor growth, invasion, and metastatic potential and shortened survival in mouse models of lung cancer and malignant melanoma [175,176]. Thus, a variety of antioxidants are recognized as tumor-promoting agents [177]. Although several rationales for this phenomenon are offered, there is a possibility that general antioxidants lacked specificity for mROS production [178,179]. Several in vivo studies have supported this opinion, in which vitamin E had no impact on the anticancer effects in a variety of cancers [174,175,176]; however, a mitochondria-targeted vitamin E analog promoted cancer cell death and had cytotoxic effects via suppression of cell energy metabolism in a xenograft breast cancer model [180]. Other murine animal models and in vitro studies have shown that spontaneous tumor metastasis and cancer cell migration was inhibited by mitochondria-targeted superoxide/ROS scavenger, mitoTEMPO [172,181].

### 5.3. Specific Regulator of mROS

As mentioned in the Introduction, many investigators have suggested that mROS was a promising therapeutic target for several types of malignant tumors [3,30,34]. There is a possibility that a variety of mitochondria-targeting compounds could be developed into novel anticancer drugs in the near future [179,182]. In addition, epitranscriptomic control of mROS is a new therapeutic strategy in a variety of cancers [3]. In vitro studies have therapeutic targets based on regulators of mROS production at the molecular level; for example, spliced x-box binding protein 1, which is associated with p53 pathway and cell metabolism [183], ubiquinol-cytochrome c reductase binding protein (UQCRB), a component of the mitochondrial complex III, and an angiogenesis-related factor [184]. Extensive studies on the specific molecular targets of mROS are necessary to establish mitochondria-targeted treatment strategies. Detailed information on the optimal range of mROS production to maximize the anticancer effects in each malignant neoplasm is important to improve outcomes in patients.

## 6. Conclusions

Mitochondrial ROS generation is an important factor influencing the anticancer activities of conventional chemotherapeutic agents. PDT may induce cell death via regulation of mitochondrial function and mROS production. In general, biological and pharmacological roles of intracellular ROS production are similar to mROS production under many pathological conditions. However, in this review, we emphasize that the mechanisms and pathological significance of mROS production is unlike that of intracellular ROS. An evaluation of mROS-specific characteristics in cancer cells is useful to develop novel treatment strategies targeting mitochondria and oxidative stress in cancer patients. In conclusion, we recommend that detailed studies should be undertaken to understand the pharmacological roles of mROS in anticancer therapy in patients with malignant tumors.

## Figures and Tables

**Figure 1 molecules-25-05252-f001:**
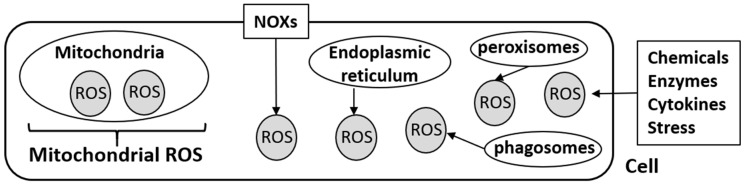
Mitochondrial reactive oxygen species (mROS) and intracellular ROS.

**Figure 2 molecules-25-05252-f002:**
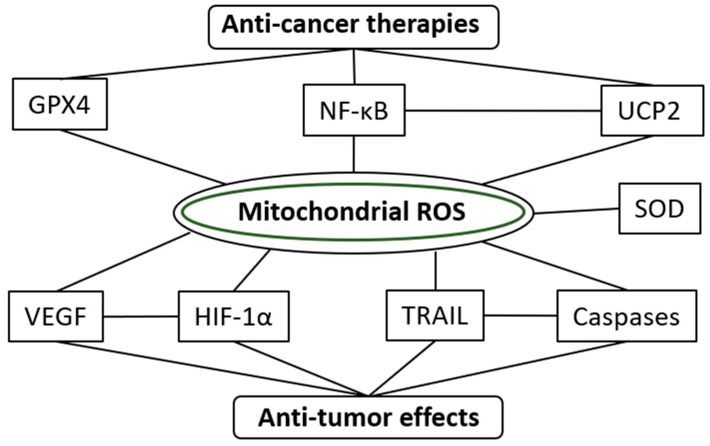
Schema of molecular mechanisms of mitochondrial ROS-related anticancer effects.

**Table 1 molecules-25-05252-t001:** Chemical compounds that affect mROS production.

Agents	Types of Malignant Tumors	Related Factor	Reference
Antimycin A	Leukemia	TRAIL	[69]
	Melanoma	TRAIL	[66]
Biguanides	Malignant lymphocytes	HIF-1α	[82]
Cisplatin	Melanoma	-	[49]
	Prostate cancer	-	[49]
	Ovarian cancer	-	[43]
Docetaxel	Colon cancer	-	[32]
FCCP	Leukemia	TRAIL	[69]
	Melanoma	TRAIL	[66]
Gemcitabine	Gallbladder cancer	UCP2, NF-κB	[60]
LGH00168	Lung cancer	NF-κB	[83]
PAM	Cervical cancer	TRAIL	[70]
Rotenone	Leukemia	TRAIL	[69]
	Melanoma	TRAIL	[66]
Temozolomide	Glioblastoma	SOD2	[65]
5-fluorouracil	Colon cancer	-	[32]

TRAIL, tumor necrosis factor-related apoptosis-induced ligand; HIF, hypoxia-inducible factor; FCCP, carbonyl cyanide p-(trifluoromethoxy) phenylhydrazone; UCP, uncoupling protein; NF-κB, nuclear factor kappa-light-chain-enhancer of activated B cells; Romo-1, reactive oxygen species modulator-1; SOD, superoxide dismutase.

**Table 2 molecules-25-05252-t002:** Natural products and photosensitizers that affect mROS production.

Agents		Type of Malignant Tumors	Related Factors	Reference
Natural products	Bi-L-RhamBet	Lung cancer	Caspases	[34]
	Curcumin	Glioblastoma	GPX4	[116]
	Matairesinol	Cervical cancer	HIF-1α, VEGF	[96]
	Tannic acid	Embryonic carcinoma	TRAIL	[110]
	Vitamin C	Embryonic carcinoma	-	[33]
Photosensitizers	Verteporfin	Glioma	-	[153]
		Astrocyte	-	[26]

GPX, glutathione peroxidase; HIF, hypoxia-inducible factor; VEGF, vascular endothelial growth factor; TRAIL, tumor necrosis factor-related apoptosis-induced ligand.

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
