# Peer review of "Pathological and Pharmacological Roles of Mitochondrial Reactive Oxygen Species in Malignant Neoplasms: Therapies Involving Chemical Compounds, Natural Products, and Photosensitizers"

_molecules, 2020, doi:10.3390/molecules25225252_

Round 1
Reviewer 1 Report
The paper "Pathological and pharmacological roles of mitochondrial reactive oxygen species in malignant neoplasms: therapies involving chemical compounds, natural products, and photosensitizers" by Miyata et al. is a well written, clear and comprehensive critical review of the latest literature concerning the emerging field of the role of mROS in malignant tumors.
Author Response
We thank the reviewers for carefully evaluating our manuscript. We are happy for your positive evaluation. In revised version of the manuscript, we modified some sentences and Table, and added Figures according to other Reviewers’ suggestion. Changes made in response to the other reviewers’ comments are highlighted in red in the revised version of the manuscript.

Reviewer 2 Report
The article reviews current knowledge on targeting mitochondrial ROS as a therapeutic strategy for cancer treatment. It puts special attention to distinguishing the effects of potential therapeutic substances on mitochondrial and cytosolic ROS levels, which is an important aspect often overlooked in the literature. The structure of the article is clear and the topic is interesting. However, before the publication I recommend the following improvements:
- Overall, the manuscript would benefit from some style editing, for example to avoid the excessive repetitions of the names of the described substances. Sometimes it is inevitable, but in many cases they could be substituted by “this substance” etc. or by modifying the sentence construction.
- The scheme illustrating the relations between mROS and regulatory pathways mentioned in the text (related to HIF1alpha, TRAIL and so on) would be a valuable summary of the presented data. There should also be added the table summarizing the data on known effects of photosensitizers on mROS, similar to Tables 1 and 2. It would present only 2 substances, but those tables are helpful in navigating the text. Alternatively, to avoid adding such short table, all three tables could be combined into one with three subsections.
- The part on vitamin C is quite long, while it presents only one study related to mitochondrial ROS. Moreover, this study does not refer to the impact of vitamin C on mROS in malignant cells, but it checked its effect on oxidative stress induced by other compound (NaF). Carcinoma cell line was only used there as a model to study the toxic effect of NaF and antioxidant potential of vitamin C. Thus, in my opinion, this part could be completely removed.
- Part 2.1.1: A549 is a non-small cell lung cancer cell line (on such cells the study from the mentioned reference was performed) and not melanoma cells, as mentioned in lines 131 and 134.
- Table 1: Romo 1 is not a chemical agent used to treat cells, but an endogenous protein, which downregulation was found to affect TRAIL-mediated apoptosis. Therefore putting it in this table is misleading.
- Lines 298-299 – “increased ROS via mitochondrial permeabilization” shall be reformulated to “increased ROS and loss of mitochondrial membrane potential“. The term “mitochondrial permeabilization” was indeed used in the referenced paper, but it is misleading, as it suggests opening of mitochondrial megachannel or disruption of mitochondrial membrane, while these aspects were not checked in the referenced study.
- Lines 410-412 – the reference shall be added. It is not obvious why exposure to sunlight is necessary to avoid damage to healthy cells, thus the reader shall be navigated to the literature explaining it.
- Last sentence of “Conclusions” – why only the immune therapy is mentioned in the final conclusion? This section shall present generalized conclusion of the whole review.
- Small typos I have spotted: line 178 “diabetic mellitus” instead of “diabetes mellitus”; line 500: “In vitro studies have r therapeutic targets”.
Author Response
Response to reviewer comments (molecules-987334)
Reviewer 2
(Reviewer’s comments)
The article reviews current knowledge on targeting mitochondrial ROS as a therapeutic strategy for cancer treatment. It puts special attention to distinguishing the effects of potential therapeutic substances on mitochondrial and cytosolic ROS levels, which is an important aspect often overlooked in the literature. The structure of the article is clear and the topic is interesting. However, before the publication I recommend the following improvements:
Response)
We thank the reviewers for carefully evaluating our manuscript. The suggestions and advices have greatly helped us in improving the manuscript. Our point-by-point responses to the comments are provided below. Changes made in response to the comments are highlighted in red in the revised version of the manuscript.
(Comments)
- Overall, the manuscript would benefit from some style editing, for example to avoid the excessive repetitions of the names of the described substances. Sometimes it is inevitable, but in many cases they could be substituted by “this substance” etc. or by modifying the sentence construction.
Response)
Thank you for important opinion. We checked all sentences in the text and modified the sentences according to your suggestion.
For examples; 1. Introduction: 1st.paragrapgh, line 3 and 2nd.paragrapgh, line2; 2.1.21 Cisplatin, line 9; 2.3 Other conventional chemotherapeutic agents, line 2; 2.4 TRAIL-related chemical compounds: 1st. paragraph, line 2 and 2nd. paragraph, line8; 2.5. Other chemical compounds: 1st. paragraph, line12; 3.1 Matairesinol: line 2; 3.4. Tannic acid: line 2; 3.5. Curcumin: line 2; 3.6 Vitamin C: line 2 and 3.
- The scheme illustrating the relations between mROS and regulatory pathways mentioned in the text (related to HIF1alpha, TRAIL and so on) would be a valuable summary of the presented data. There should also be added the table summarizing the data on known effects of photosensitizers on mROS, similar to Tables 1 and 2. It would present only 2 substances, but those tables are helpful in navigating the text. Alternatively, to avoid adding such short table, all three tables could be combined into one with three subsections.
Response)
Thank you for your excellent suggestions. According to your opinion, we made new Figure about regulatory mechanism of mROS-related anti-cancer effects (new Figure 2). We believe that this Figure helps to understand our review about pathological and pharmacological roles of mROS.
In addition, we modified Table2 about photosensitizers that affect mROS production (Table 2) based on your suggestion. However, in this time, we showed the summary of natural products and photosensitizers that affect mROS production as Table 2. We believe that these contents are most newly and highly unique ones in this review. Therefore, we want to show such information as independent Table.
- The part on vitamin C is quite long, while it presents only one study related to mitochondrial ROS. Moreover, this study does not refer to the impact of vitamin C on mROS in malignant cells, but it checked its effect on oxidative stress induced by other compound (NaF). Carcinoma cell line was only used there as a model to study the toxic effect of NaF and antioxidant potential of vitamin C. Thus, in my opinion, this part could be completely removed.
Response)
We agree with your opinion. Therefore, at first, we thought to delete all contents about vitamin C. However, vitamin C is one of most well-known natural products and anti-oxidants. In addition, the information that pharmacological effect of mROS by vitamin C in embryonic carcinoma is important for the readers. Therefore, we modified and simplified the contents about vitamin C, as well as other natural products (3.6 Vitamin C).
- Part 2.1.1: A549 is a non-small cell lung cancer cell line (on such cells the study from the mentioned reference was performed) and not melanoma cells, as mentioned in lines 131 and 134.
Response)
We are sorry for simple mistakes. In the revised version of the manuscript, A459 cells were showed as a non-small cell lung cancer cell line (2.1.1 Cisplatin: 1st.paragrapgh, lines 1 and 4 – 5).
- Table 1: Romo 1 is not a chemical agent used to treat cells, but an endogenous protein, which downregulation was found to affect TRAIL-mediated apoptosis. Therefore putting it in this table is misleading.
Response)
We agree with your opinion. Therefore, we deleted the information on Romo 1 from Table 1. On the other hand, main content of the text is not changed because information on relationship between Romo 1 and mROS production was showed in the 2.4 TRAIL-related chemical compounds (2nd. paragraph).
- Lines 298-299 – “increased ROS via mitochondrial permeabilization” shall be reformulated to “increased ROS and loss of mitochondrial membrane potential“. The term “mitochondrial permeabilization” was indeed used in the referenced paper, but it is misleading, as it suggests opening of mitochondrial megachannel or disruption of mitochondrial membrane, while these aspects were not checked in the referenced study.
Response)
Thank you for important suggestion. We agree with your opinion. Therefore, we modified the sentence according to your opinion (3.2 Pancratistatin: lines 5 – 6).
- Lines 410-412 – the reference shall be added. It is not obvious why exposure to sunlight is necessary to avoid damage to healthy cells, thus the reader shall be navigated to the literature explaining it.
Response)
We agree with your opinion. This sentence is not associated with main theme of this review. Therefore, we deleted this sentence to avoid misunderstanding in the revised version of the manuscript (4.1. Photosensitizers and mROS).
- Last sentence of “Conclusions” – why only the immune therapy is mentioned in the final conclusion? This section shall present generalized conclusion of the whole review.
Response)
We agree with your opinion. In the revised version of the manuscript, we modified the sentence (immune therapy → anti-cancer therapy) (6. Conclusions: last sentence).
- Small typos I have spotted: line 178 “diabetic mellitus” instead of “diabetes mellitus”; line 500: “In vitrostudies have r therapeutic targets”.
(Response)
We are sorry for simple mistakes. We modified them and checked all sentences in the text of revised version of the manuscript (2.2 Taxane: 2nd. paragraph, last sentence).

Reviewer 3 Report
The development of patient-specific therapeutic approaches for malignant tumors is of great importance. In this context mROS may serve as an important characteristic and predictive marker of malignant tumors in development of efficient treatment strategies. The review “Pathological and pharmacological roles of mitochondrial reactive oxygen species in malignant neoplasms: therapies involving chemical compounds, natural products, and photosensitizers” is written moderately concisely, and the accents are very well placed.
In the introduction the authors are exceptionally correct in discussing the role of various sources of ROS, they do not underestimate NADPH oxidase and discuss both intracellular ROS and mROS.
Further, the authors, citing corresponding literature, convincingly show that anti-cancer effects of some chemicals (including Cisplatin and Taxane) can be enhanced by adequately controlling mROS production. In Table 1 the authors probably mentioned most anti-cancer chemical compounds affecting mROS production. Anti-cancer natural products that affect mROS production are presented in Table 2. Biological role of mtROS in cancer treatment with photosensitizers has been analyzed also.
And finally, an important structural part is section 5, Future perspectives on the pharmacological roles of mROS in cancer treatments.
A good review was recently published on the topic of mROS (Front. Oncol., 28 February 2020 | https://doi.org/10.3389/fonc.2020.00256), but the manuscript of Yasuyoshi Miyata et al. adds a plenty of data about cases where targeting of Mitochondrial-mediated processes enhanced anti-cancer efficacy of compounds.
ROS activating pathways are critical for cancer progression. Undoubtedly, the authors have carried out a fundamental analysis of mROS mediated antitumor effects of chemical compounds, natural products, and photosensitizers, but the paper lacks some visual diagram summarizing the research of mROS, approaches for controlling mROS in line with anti-cancer therapy, and why the mechanisms and pathological significance of mROS production is unlike that of intracellular ROS.
Author Response
Response to reviewer comments (molecules-987334)
Reviewer 3
(Reviewer’s comment)
The development of patient-specific therapeutic approaches for malignant tumors is of great importance. In this context mROS may serve as an important characteristic and predictive marker of malignant tumors in development of efficient treatment strategies. The review “Pathological and pharmacological roles of mitochondrial reactive oxygen species in malignant neoplasms: therapies involving chemical compounds, natural products, and photosensitizers” is written moderately concisely, and the accents are very well placed.
In the introduction the authors are exceptionally correct in discussing the role of various sources of ROS, they do not underestimate NADPH oxidase and discuss both intracellular ROS and mROS.
Further, the authors, citing corresponding literature, convincingly show that anti-cancer effects of some chemicals (including Cisplatin and Taxane) can be enhanced by adequately controlling mROS production. In Table 1 the authors probably mentioned most anti-cancer chemical compounds affecting mROS production. Anti-cancer natural products that affect mROS production are presented in Table 2. Biological role of mtROS in cancer treatment with photosensitizers has been analyzed also.
And finally, an important structural part is section 5, Future perspectives on the pharmacological roles of mROS in cancer treatments.
A good review was recently published on the topic of mROS (Front. Oncol., 28 February 2020 | https://doi.org/10.3389/fonc.2020.00256), but the manuscript of Yasuyoshi Miyata et al. adds a plenty of data about cases where targeting of Mitochondrial-mediated processes enhanced anti-cancer efficacy of compounds.
ROS activating pathways are critical for cancer progression. Undoubtedly, the authors have carried out a fundamental analysis of mROS mediated antitumor effects of chemical compounds, natural products, and photosensitizers, but the paper lacks some visual diagram summarizing the research of mROS, approaches for controlling mROS in line with anti-cancer therapy, and why the mechanisms and pathological significance of mROS production is unlike that of intracellular ROS.
Response)
We thank the reviewers for carefully evaluating our manuscript. We are happy for your positive evaluation. In revised version of the manuscript, we added the new Figures about the difference of productions between mROS and intracellular ROS (new Figure 1) and the pathological significance of mROS in anti-cancer therapy (new Figure 2) based on your opinion.
In addition, we modified some sentences and Table according to other Reviewer’s suggestion. Changes made in response to the other reviewers’ comments are highlighted in red in the revised version of the manuscript.
